# Immunotherapy in Acute Leukemias: Past Success Paves the Way for Future Progress

**DOI:** 10.3390/cancers15164137

**Published:** 2023-08-17

**Authors:** Adel Chergui, John L. Reagan

**Affiliations:** Division of Hematology and Oncology, Rhode Island Hospital, Alpert Medical School of Brown University, Providence, RI 02903, USA; adel_chergui@brown.edu

**Keywords:** acute leukemia, immunotherapy, allogeneic stem cell transplant, immune checkpoint inhibitors, antibodies

## Abstract

**Simple Summary:**

Recent breakthroughs in cancer therapeutics have occurred through the recognition of the immune system’s role in cancer cell eradication and surveillance. While these breakthroughs have primarily occurred in solid tumors, the role of immunotherapy as an anti-cancer treatment is best described through decades of work in allogeneic hematopoietic stem cell transplantation (allo-HSCT) for acute leukemias. Herein we review the history and mechanisms behind allo-HSCT and discuss how lessons learned through transplants and cancer evasion of the host immune system can be applied toward acute leukemias to provide future therapies.

**Abstract:**

Immunotherapy as a cancer treatment modality has undergone recent widespread proliferation across all cancer types, especially amongst patients with solid tumors. However, the longest tenured immunotherapy approach to cancer is allogeneic stem cell transplantation (allo-SCT) for two hematologic malignancies: acute myeloid and acute lymphoid leukemia (AML and ALL, respectively). While allo-SCT remains a standard of care for eligible patients, recent advances/applications of monoclonal antibodies, immune checkpoint inhibitors, bispecific T-cell engagers (BiTEs), and CAR T-cell therapy are changing the treatment landscape for these acute leukemias by either direct to tumor immune targeting or through decreased toxicities that expand patient eligibility. Pre-clinical data and clinical trials have shown promising results for novel immunotherapies in acute leukemia, and multiple ongoing trials are investigating these novel approaches. While there have been promising results with these approaches, particularly in the relapsed/refractory setting, there remain challenges in optimizing the use of these therapies, such as managing cytokine release syndrome and other immune-related toxicities. Immunotherapy is a rapidly evolving field in the treatment of acute leukemia and has the potential to significantly impact the management of both AML and ALL. This review highlights the history of immunotherapy in the treatment of acute leukemias, the evolution of immunotherapy into more targeted approaches, the potential benefits and limitations of different immune targeting approaches, and ongoing research and development in the field.

## 1. Introduction

In recent decades, significant advances have been made in the field of immunotherapy revolutionizing the treatment landscape for various malignancies including acute leukemias. Among the earliest and most impactful therapeutic modalities explored was allogeneic hematopoietic stem cell transplantation (allo-HSCT), which utilized the graft-versus-leukemia effect to treat acute leukemias. This paper presents a historical perspective on the development of immunotherapy in acute leukemias, starting first with the pivotal role played by allo-HSCT then exploring the emerging and promising future directions for immunotherapeutic treatments in acute leukemias as well as limitations of this therapeutic strategy. Select mechanisms by which immunotherapy targets acute lymphocytic leukemia (ALL, Figure 1) and acute myeloid leukemia (AML, Figure 2) are discussed.

## 2. Early Immunotherapy and Allogenic Hematopoietic Stem Cell Transplantation

Dr. William B. Coley, often referred to as the “father of immunotherapy”, laid the foundation for modern cancer immunotherapies. His work began after observing the remission of sarcoma in a patient who had developed a concurrent bacterial skin infection [1]. Dr. Coley formulated a mixture of killed bacteria, known as “Coley’s Toxins”, and began to experiment with its therapeutic potential in sarcomas. Although met with skepticism in his time, his work eventually paved the way for understanding the intricate relationship between the immune system and malignancy [2]. 

The concept of using the immune system to combat acute leukemias further evolved with the advent of allogeneic hematopoietic stem cell transplantation (allo-HSCT). The pioneering work of E. Donnall Thomas in 1957 demonstrated the first use of allo-HSCT in a case series of six patients [3]. This landmark study provided a foundation for subsequent investigations exploring the role of allo-HSCT in hematological malignancies [4,5]. The term adoptive immunotherapy was first coined by Mathe et al. in 1965 in recognition that immunocompetent allogeneic cells could eradicate host leukemic cells. Perhaps most remarkable was the development of this concept prior to the discovery of human antigen typing or even recognition of graft-versus-host disease apart from being described as a “secondary syndrome induced by the homograft” [6]. 

Subsequent studies confirmed that the curative potential of allo-HSCT was not solely attributed to the replacement of the diseased marrow but also to an immunological phenomenon known as the graft-versus-leukemia (GVL) effect [7]. This phenomenon, which was linked to the development of graft-versus-host disease (GVHD), was characterized as the recognition and eradication of residual leukemia cells by the donor-derived immune cells leading to improved outcomes in both Acute Lymphoblastic Leukemia (ALL) and acute myeloid leukemia (AML) [8]. The notion of adoptive immunotherapy as an anti-leukemia treatment mechanism was a paradigm shift. Initially, the beneficial effect of allo-HSCT was thought to be from high-dose chemotherapy and/or radiation directly killing residual host leukemia cells. The donor cells simply served as a source for rescue hematopoiesis. These early studies linking GVL to GVHD laid the foundation for subsequent research that explored various strategies to improve the therapeutic role of allo-HSCT, including the use of T-cell depletion, donor lymphocyte infusions, and post-transplant immunosuppression optimization [9]. 

The role of T-cells in the graft-versus-leukemia effect became evident in 1990 when research showed an increased risk of relapse after T-cell depletion in various hematological malignancies. Notably, the risk of relapse in T-cell depleted transplants was the most pronounced in CML, less in acute myeloid leukemia (AML), and lowest in acute lymphoblastic leukemia (ALL) [10]. More recently, studies have challenged the notion of T cells as the main driver of GVL in AML and less important for ALL. In an EBMT study of 48,000 allo-HSCTs, the strength of the GVHD/GVL correlation was strong in ALL but less prevalent amongst AML patients. These findings suggest that the GVL effect in AML might be less T-cell-mediated than previously thought and may instead rely on other immune effector cells such as natural killer (NK) cells or B-cells [11]. The questionable role of T-cells in the graft-versus-leukemia response in AML is further supported by a recent retrospective review of 266 AML patients who received T-cell-depleted allo-HSCT in CR1 or CR2. Survival, both overall and disease free, was compared favorably with a cohort of unmodified allo-HSCT patients but with less GVHD in the T-cell-depleted group [12]. Conversely, the correlation between both acute and chronic GVHD in ALL was confirmed in a CIBMTR analysis of 5215 ALL patients who underwent allo-HSCT [13].

The contribution of B-cells to the GVL effect has been reported in a group of high-risk AML patients with durable remissions and AML-specific cytotoxic antibodies. The authors were able to detect monoclonal antibodies directed against AML cell surface antigens that were leukemia-specific and not on normal hematopoietic or endothelial cells. Many of the antibodies recognized the U5 snRNP200 complex which is a spliceosome that is found intracellularly in normal cells but in leukemic blasts present on the cell surface. The investigators were able to show that these U5 snRNP200 complex-specific antibodies could induce leukemic death through an Fc receptor-dependent mechanism and could inhibit leukemic growth in a murine model further underscoring the possible contribution of antibodies to the GVL effect [14]. 

Despite the beneficial effects, allo-HSCT it is still not universally available for all acute leukemia patients. Previously, a lack of a suitable donor was a major barrier to a transplant which has now been largely overcome through improved outcomes in mismatched grafts and cord blood transplants. Now, the limitation is primarily based on the patient’s age and co-morbid conditions which can be evaluated by the hematopoietic cell transplantation co-morbidity index [15]. Although initially validated in younger patients less than 60 years, the HCT-CI can also be used for patients over 60 to determine fitness for allo-HSCT [16]. It provides an additional tool for the provider to consider when evaluating an older patient’s fitness for allo-HSCT as a possible therapy. 

Relapse remains the most common cause of death for allo-HSCT patients and is largely mediated through antigen recognition by the allogeneic cells, either through HLA loss following mismatched transplants, downregulation of major histocompatibility genes in leukemic blasts, and an increase in immune inhibitory checkpoint markers on leukemic blasts and donor T-cells [17,18,19,20,21] (Figure 3).

## 3. Immune Checkpoint Blockade

Building on the role of T-cells as a therapeutic target in malignancies, immune checkpoint blockade has revolutionized the treatment landscape for many solid tumors. Expanding the repertoire of immunotherapeutic approaches in leukemia, immune checkpoint inhibitors have emerged as a promising target to augment the graft-versus-leukemia effect following allogenic transplant. Disease relapse following allogenic transplant in leukemias is associated with poor survival with limited treatment options [22]. The mechanism of relapse following allo-HSCT is partly believed to be driven by the deregulation of multiple inhibitory ligands at relapse such as programmed death 1 (PD-1), cytotoxic T-lymphocyte associated protein 4 (CTLA-4) receptors and their associated ligands PD-L1, PD-L2, B7-1, and B7-2, which enables immune escape through T-cell exhaustion leading to disease progression [21].

The first study to show the effectiveness of checkpoint blockade in relapsed hematological malignancies after allo-HSCT was a phase I/Ib clinical trial, which examined the use of CTLA-4 inhibition, specifically using ipilimumab, on patients with relapsed hematological malignancies following allo-HSCT. Twenty-eight patients were enrolled of which 14 had refractory or relapsed AML and MDS. The trial demonstrated that 5 out of the patients with myeloid malignancies managed to achieve a complete response. Nevertheless, the study also reported significant adverse events. Specifically, 6 patients (21%) experienced grade ≥3 immune-related adverse events (irAEs), including 1 fatality. Furthermore, 4 patients (14%) developed graft-versus-host disease (GVHD), which prevented further administration of ipilimumab [23]. Another retrospective multi-center study examined 21 patients with hematological malignancies other than Hodgkin’s Lymphoma (HL) who received checkpoint inhibitors. Of those, 14 had leukemias (12 MD/AML and 2 ALL) that had relapsed after allo-HSCT. The patients received either nivolumab or ipilimumab alone, a combination of both, or a combination of nivolumab with donor lymphocyte infusion. The ORR was 43% (3 CR, 6 PR), with higher response rates observed in patients receiving nivolumab plus DLI (ORR = 80%) compared to patients receiving nivolumab alone (ORR = 40%) or ipilimumab alone (ORR = 20%) [24]. 

Later studies have shown the limited use of checkpoint blockade in relapsed myeloid malignancies after allo-HSCT. In a prospective study, David et al. in 2020 showed minimal activity of nivolumab in relapsed myeloid malignancies after allo-HSCT (ORR = 22%) and modest activity in lymphoid malignancy (ORR = 44%) [25]. In another study, examining the role of pembrolizumab in relapsed hematological malignancies after allo-HSCT showed no response in myeloid malignancies [26]. Furthermore, the use of checkpoint blockade in the post-transplant setting may be complicated by immune-related adverse events (irAEs). A phase 1 study conducted by Wang et al. examining the role of checkpoint blockade with nivolumab as a maintenance therapy following allo-HSCT was terminated due to irAEs. Of the four patients treated, all developed irAEs, of which two were serious including a grade 4 neutropenia and grade 3 encephalopathy [27].

Outside of the post-transplant setting, combination strategies involving immune checkpoint inhibitors with hypomethylating agents (HMAs) are being actively explored both in the front-line and relapsed/refractory setting to further enhance therapeutic efficacy in AML. The rationale of combining hypomethylating agents with checkpoint blockade is based on upregulation of genes expressing checkpoint blockade (PD-L1, PD-L2, CTLA4) following exposure to HMAs [28]. An initial study that explored the use of nivolumab in combination with azacitidine and a later study that used a combination of nivolumab and ipilimumab with azacitidine in relapsed/refractory AML had modest results [29,30]. More recent studies established the safety of the combination of decitabine and pembrolizumab in refractory and relapsed AML while additional studies establishing efficacy are ongoing [31]. The combination of PD-L1 blockade with hypomethylating agents in the front-line setting was examined in a study by Zeidan et al. which combined the PD-L1 inhibitor durvalumab in combination with azacitidine. Unfortunately, this study did not show any clinical efficacy in comparison to azacitidine alone [32]. Currently, other immune checkpoint targets such as TIM-3 in combination with HMAs are being investigated in early clinical trials showing promising early results [33]. Other checkpoint receptors such as LAG-3 and TIGIT are being explored as potential targets but thus far have been explored in pre-clinical studies [19,20,34].

Immune checkpoint blockade has also been explored in combination with cytotoxic chemotherapy for acute myeloid leukemia. The scientific rationale for this combination is based on pre-clinical studies showing that cytarabine suppresses PD-1 on myeloblasts thereby enhancing their killing by cytotoxic T-cells and thus restoring immune surveillance [35]. Cytotoxic chemotherapy also leads to the release of neo-antigens which may help activate cytotoxic T-cells further enhancing anti-tumor activity. Additionally, PD-1 expression is seen more frequently in patients with relapsed AML [20]. Using this rationale, a single-arm phase II study examined the use of idarubicin, cytarabine, and nivolumab with newly diagnosed AML or high-risk MDS. Results from the study showed that the addition of nivolumab was safe, however, median overall survival and response rate were not much improved over receiving intensive chemotherapy alone. Furthermore, the study showed that CD4^+^ T-effector cells of non-responders displayed an exhausted phenotype which may be an avenue for future targets [36]. In another trial, investigators looked at the use of pembrolizumab after high-dose cytarabine (HiDAC) in relapsed/refractory AML which further established the safety of using immune checkpoint blockade in combination with cytotoxic chemotherapy and showed clinical activity which requires further investigation [37]. 

Anti-CD47 monoclonal antibody therapy represents an exciting frontier in the immunotherapy of acute leukemias and provides a shift away from T-cell-directed immune approaches. CD47 is broadly expressed on normal cells. It acts as a ‘do not eat me’ signal by binding to the signal-regulatory protein alpha (SIRPα) on macrophages. By overexpressing CD47, leukemic blasts can evade phagocytosis facilitating their survival and proliferation. CD47/SIRPα therefore presents a promising new target in immune checkpoint blockade [38]. In 2009, Majeti and colleagues first demonstrated that blocking CD47 on AML cells leads to an increase in phagocytosis [39]. Subsequent pre-clinical research showed that anti-CD47 monoclonal antibodies can effectively facilitate the destruction of AML cells by macrophages in xenograft models [40]. Magrolimab, a CD47 monoclonal antibody, has been studied in early phase clinical trials in combination with azacitidine and venetoclax and was found to have an acceptable safety profile with encouraging response rates [41]. These early results have led to an ongoing phase III trial (ENHANCE-3) to study its efficacy in AML (Table 1) [42]. 

The role of immune checkpoint blockade in B-ALL remains limited due to a lack of pre-clinical evidence showing efficacy. However, there is evidence that checkpoint blockade has a role to play in resistance to the bispecific T-cell engager targeting CD3 and CD19 and CAR T-cell blinatumomab [43]. There is also evidence that an exhausted CD4 T-cell phenotype can predict relapse in adult B-ALL [44,45]. Pre-clinical studies have shown some promising results using combination therapy with PD-L1 blockade [46]. Currently, there is an ongoing trial examining the combination of blinatumomab and nivolumab with and without ipilimumab in relapsed or refractory CD19^+^ B-ALL [47].

**Table 1 cancers-15-04137-t001:** Select immunotherapy agents in late-stage clinical trials or FDA-approved development phases for acute leukemias.

Agent	Target	Malignancy	Outcomes	Trial/Development Phase	Reference
**Monoclonal Antibodies**					
Magrolimab	CD47/SIPRα	AML, first-line	Trial ongoing	Phase III clinical trial	[42]
Rituximab	CD20	ALL	Ritux + chemo vs. chemo alone: 2-year event-free survival 65% vs. 52% (*p* = 0.04)	FDA approved	[48,49]
**Immune Cell Engagers**					
Blinatumomab	CD19/CD3	Relapsed and refractory ALL	Blina vs. chemo: median overall survival 7.7 months vs. 4 months (*p* = 0.01)	FDA approved	[50]
		MRD + ALL	MRD after 1 cycle of blina: 80%, 95% CI [71–87]	FDA approved	[51]
		Newly diagnosed Ph + ALL	Dasatinib induction followed by blina + dasatinib consolidation: 18-month survival 95%, MRD response rate 52%	Phase II clinical trial	[52]
**Chimeric Antigen T-Cell Therapy (CAR-T)**					
Tisagenlecleucel	CD19	ALL, relapsed/refractory	Three-month complete remission rate: 81%, 95% CI [69–91]	FDA approved	[53]
Brexucabtagene autoleucel	CD19	B-ALL, relapsed/refractory	Complete remission rate: 71%, 95% CI [57–82]	FDA approved	[54]

## 4. Monoclonal Abs

Monoclonal antibody therapy has emerged as a critical element of the immunotherapeutic landscape in hematological malignancies. This section will explore two key instances of monoclonal antibody therapy in leukemias—anti-CD20 therapy in B-ALL and anti-CD38 in T-ALL. Both therapies illustrate the potential of targeting specific antigens on the surface of leukemic cells to enhance the immune response to eliminate leukemic cells. 

### 4.1. Anti-CD20

CD20, a transmembrane protein, is primarily expressed on the surface of mature B- cells and most B-cell malignancies, including B-ALL. Rituximab, a chimeric anti-CD20 monoclonal antibody, was initially used with tremendous efficacy in non-Hodgkin’s lymphoma (NHL) [55]. The mechanisms by which anti-CD20 antibodies eliminate CD20^+^ cells include antibody-dependent cellular cytotoxicity (ADCC), antibody-dependent cellular phagocytosis, complement-dependent cytotoxicity (CDC), and apoptosis [56]. Recognizing its success in NHL, as well as the observation that increased expression of CD20 cells in adults with B-ALL conferred a poor prognosis led to further trials exploring its use in B-ALL [57,58]. 

Rituximab use in Ph- B-ALL was first studied in a single-center phase II trial by Thomas et al. in 2010 which showed improved outcomes with the addition of rituximab to hyper-CVAD in patients with Burkitt’s lymphoma or B-ALL [48]. Following this, the randomized phase III GRAALL-R 2005 study published in 2016 showed a significant survival advantage when rituximab was added to standard chemotherapy for patients with CD20^+^ Ph-negative ALL [49]. This has led to the inclusion of rituximab as standard therapy for any CD20^+^ B-ALL (Table 1). 

### 4.2. Anti-CD38

The development of immunotherapy for T-cell ALL (T-ALL) has lagged development for B-ALL for several reasons including the heterogeneity of cell surface expression in T-cell lymphoblasts. Pre-clinical studies have shown that T-ALL blasts have cell surface expression of CD38, a transmembrane receptor expressed on thymocytes, activated T-cells, terminally differentiated B-cells but minimally expressed in normal myeloid and lymphoid cells, making it a promising therapeutic target [59]. 

Daratumumab, an anti-CD38 monoclonal antibody, which was studied and approved for the front-line treatment of multiple myeloma [60], has shown pre-clinical efficacy in T-ALL xenograft models [59,61]. Small studies in patients with T-ALL have also shown efficacy in eradicating MRD positivity in pediatric patients with T-ALL [62]. A recent retrospective study looking at daratumumab in addition to and without chemotherapy in relapsed or refractory T-ALL further supports that there may be therapeutic effect in targeting CD38 in relapsed or refractory T-ALL [63]. Given these encouraging studies, the DELPHINUS study, a phase II clinical trial in pediatric, adolescent, and young adults, was conducted to assess the efficacy and safety of daratumumab in relapsed refractory T-ALL. Early results from the study showed improved response rates when daratumumab is added to chemotherapy versus chemotherapy alone with 83% complete response rates in children and 60% in adults with relapsed T-ALL [64]. Combination therapy with anti-CD47 is also being considered for future therapy [65].

## 5. Immune Cell Engagers (ICE)

Among the emerging immunotherapeutic strategies in acute leukemias, immune cell engagers (ICEs) have garnered significant attention. These antibody-derived molecules have the unique capability to bind various populations of normal immune cells, including T-cells, NK cells, and macrophages, as well as cancer cells, redirecting the cytotoxic activity of effector cells against the tumor.

In the context of acute lymphoblastic leukemia (ALL), one of the pioneering ICEs is blinatumomab, a bispecific T-cell engager (BiTE) that targets both CD3 and CD19. By bridging T-cells and B-lymphoid targets, blinatumomab effectively redirects T-cell cytotoxicity against leukemic cells. Its success in relapsed/refractory (R/R) Philadelphia chromosome-negative B-cell ALL (Ph^−^ B-ALL) has been demonstrated in the pivotal TOWER trial, a phase III study where continuous infusion of blinatumomab showed superior overall response rates and overall survival compared to standard of care [50]. Encouragingly, subsequent investigations revealed similar efficacy in relapsed and refractory Philadelphia chromosome-positive ALL (Ph^+^ B-ALL) and promising results in B-ALL with persistent minimal residual disease (MRD), leading to approval of blinatumomab in these indications [51]. Recent phase III studies in pediatric, adolescent, and young adults with B-ALL in second remission further support the use of blinatumomab as a consolidation strategy prior to allogeneic hematopoietic stem cell transplantation (allo-HSCT, Table 1) [66,67]. 

Building on the success of blinatumomab in the refractory setting, the D-ALBA study examined the use of a first-line combination of dasatinib and corticosteroid induction in Philadelphia-positive B-ALL followed by blinatumomab consolidation [52,68]. This study showed high rates of molecular response (29% after dasatinib induction, followed by 60% after blinatumomab consolidation). Another study by Jabbour et al. looked at blinatumomab used in induction in combination with ponatinib in both newly diagnosed and relapsed Ph^+^ B-ALL. Results from this phase II trial showed high rates of complete molecular response (87% in newly diagnosed, 92% in relapsed/refractory Ph^+^ B-ALL). This strategy of combining ICEs with TKIs in Ph^+^ B-ALL has led to higher rates of molecular response and survival leading to the discussion on whether allo-SCT is needed in first remission in Ph^+^ B-ALL [69].

ICEs are associated with specific adverse events, including cytokine release syndrome (CRS) and immune effector cell-associated neurotoxicity syndrome (ICANS). However, blinatumomab has shown a better safety profile than chemotherapy, with reduced incidences of mucositis, cytopenias, and febrile neutropenia [50,66,67]. Fortunately, blinatumomab has a short half-life (2 h) and adverse effects can be rapidly reversed with cessation of the infusion and steroid administration. 

The use of ICEs has witnessed a remarkable expansion within the realm of AML and other myeloid malignancies. These ICEs include a variety of therapeutics such as bispecific T-cell engagers (BiTEs), dual-affinity retargeting (DART) antibodies, and trispecific killer engagers (TriKEs), all designed to stimulate immune responses against various cell surface targets on leukemic blasts.

AMG330 (eluvixtamab) is a human BiTE that targets CD3 and CD33 redirects T-cell cytotoxicity towards CD33-positive AML cells. Preliminary results from studies involving heavily pre-treated AML patients have shown some activity, with higher response rates observed in patients with low tumor burden [70]. Following a similar strategy, AMG673, another CD33-targeted BiTE, has been developed with an extended half-life allowing for weekly dosing. Its phase I trial showed reductions in bone marrow blasts in some patients, with the most common treatment-related adverse event being CRS [71].

Vibecotamab, another BiTE targeting CD3 and CD123, had shown activity in heavily pre-treated refractory/relapsed AML patients. Patients who responded were found to have lower pre-therapy disease burden [72]. Flotetuzumab, a dual-affinity retargeting antibody (DART), targets CD3 and CD123 has also shown promise in the treatment of relapsed/refractory AML, particularly in patients with primary induction failure or early relapse. Notably, flotetuzumab has demonstrated encouraging results in TP53-mutated AML, and a bone marrow 10-gene expression signature has been identified as a predictor of response to flotetuzumab, correlating with immune cell infiltration [73]. 

TriKEs, another class of ICEs, are under evaluation. These novel constructs are capable of engaging natural killer (NK) cells to mediate the killing of AML cell lines. They contain a wild-type IL-15 moiety and domains against CD16 and AML-specific antigens. GTB3550, a first-in-class TriKE, was evaluated among a small group of patients with R/R AML, revealing reproducible NK cell activity in all patients and suggesting that TriKEs could be a promising alternative to CD33-targeted agents [74].

## 6. Cellular-Based Therapies

The development and clinical application of ICEs have ushered in a new era in the treatment of acute leukemias, demonstrating unprecedented response rates and improved survival outcomes. Another groundbreaking development that is changing the paradigm of acute leukemia management are cellular-based therapies such as chimeric antigen receptor T-cell (CAR-T) therapy. CAR-T are engineered T-cells that act by bringing together an extra-cellular domain which recognizes a tumor antigen independent of major histocompatibility complex proteins. Its use was pioneered in B-ALL and is now making inroads into the treatment of other acute leukemias. In the following section, we will explore the evolution, clinical efficacy, and challenges of CAR-T therapy in the management of acute leukemias.

The first successful CAR-T target was CD19 which was developed for CD19 expressing malignancies including B-ALL. Tisagenlecleucel, an autologous anti-CD19/4-1BB/CD3z CAR-T, was studied in the phase I/IIa ELIANA trial in which pediatric and young adult patients with CD19^+^ relapsed or refractory B-ALL were treated. Results showed high rates of initial response in relapsed patients and long-term remission in some [53,75,76]. Following the results of this study, FDA approval was given for use in treatment of children and young adults up to 25 years old with B-ALL in second relapse, in first relapse after allo-HSCT, or refractory to induction. Brexucabtagene autoleucel, an autologous CD19/CD28/CD3z CAR-T, showed equally robust clinical data in the phase II ZUMA-3 trial which evaluated its safety and efficacy in relapsed/refractory B-ALL [54]. Long-term data from ZUMA-3 shows complete remission among treated patients was 71% and OS was 25.4 months [77]. A systematic review and meta-analysis by Elsallab et al. in 2023 showed that the overall response rate was 76% and the median overall survival of currently available autologous anti-CD19 CAR-T in relapsed/refractory B-ALL is 36.2 months (Table 1) [78]. 

One of the challenges of autologous CAR-T encountered in previous trials was that certain patients could not be included due to manufacturing failure, disease progression, or severe adverse events occurring during the manufacturing process. Allogenic CAR-T or “off-the-shelf” CAR-T have been developed to reduce time in CAR T-cell infusion to overcome those challenges. UCART-19, allogeneic 4-1BB/CD3z CAR T-cells manufactured from non-HLA compatible donors, were investigated in two phase I clinical trials (CALM trial in adults, PALL in pediatrics) in patients with relapsed/refractory B-ALL. These trials showed a favorable safety profile and some anti-leukemic activity [79,80].

While CAR-T remains a promising therapeutic option for relapsed/refractory B-ALL, its use is limited by significant toxicities, namely, cytokine release syndrome (CRS) and immune effector cell-associated neurotoxicity syndrome (ICANS). These toxicities have been reported in higher rates in ALL patients as compared to other B-cell malignancies, especially in adults. Factors associated with higher severity CRS include higher T-cell dose, and pre-treatment bone marrow burden and risk factors for ICANS include bone marrow disease burden, cyclophosphamide and fludarabine lymphodepletion, presence of any pre-existing neurologic comorbidity and CD8^+^ CAR T-cell dose and peak expansion [81]. To overcome the toxicities of anti-CD19 CAR-T, a “fast off” second-generation CD19/4-1/CD3z CAR-T called Obecabtagene autoleucel was developed and studied in the ALLCAR19 trial [82]. Obecabtagene autoleucel quickly dissociates from CD19 which mimics physiological T-cell activation leading to lower adverse events. The results of the trial showed 85% of studied patients (*n* = 20) achieved MRD-CR without any grade 3 adverse events. These results led to the ongoing FELIX trial which is a phase II trial which is using this construct in adult patients with relapsed/refractory B-ALL (NCT04404660) [83]. Other strategies to reduce CRS and ICANS include prophylactic dexamethasone and tocilizumab administration making toxicities from treatment more tolerable and safer [84].

In contrast to the clinical success of CAR-T-cell therapy in ALL, the development of CAR T-cells in myeloid malignancies remains challenging. The major challenge with the development of CAR-T in myeloid malignancies is that myeloid leukemia cells share cell surface markers with normal hematopoietic cells which may lead to prolonged cytopenias which is a much more difficult problem to deal with compared to the lymphopenia seen in targeting CD19 in B-ALL [85]. Multiple CAR-T-cells therapies are in development in early phase clinical trials targeting CD33, CD123, and NKG2D. 

In the ongoing exploration of cellular-based treatments for AML, natural killer (NK) cells have garnered attention. NK cells are innate lymphoid cells known for their inherent ability to recognize and destroy tumor cells without prior sensitization. Their early promise was initially showcased in the context of T-cell-depleted mismatched-donor allo-HSCT. Researchers observed that donor vs. recipient NK cell alloreactivity avoids relapse and graft rejection in patients with AML [86]. Subsequent studies showed that infusions of haploidentical NK cells following non-myeloablative chemotherapy can induce complete remissions in patients with AML without the use of transplant [87]. These observations underscored the potential anti-leukemic properties of NK cells, suggesting their therapeutic relevance in AML. The limitations of NK-based therapies are due to the limited in vivo persistence and limitations in procurement and processing [88]. 

With the advancement of genetic engineering, researchers have begun to explore the potential of genetically modified NK cells to further harness and amplify their anti-leukemic activity specifically against AML targets. Preliminary results from clinical trials using CAR-NK cells and other therapeutic NK cell products are indeed promising, however, they are currently in the early phases of clinical trials and have yet to achieve FDA approval [88].

## 7. Conclusions

The spotlight on T-cells in the realm of immunotherapy for acute leukemias is well deserved, given their crucial role in graft-versus-leukemia effect. However, as highlighted throughout this review, there is growing recognition that additional targets beyond T-cells hold significant promise. From immune checkpoint inhibitors targeting PDL-1 to monoclonal antibodies against CD20, CD38, and CD47, as well as the emerging field of immune cell engagers and genetically modified cell therapies, new avenues for effective treatment strategies are being explored. While T-cells have paved the way, the horizon of immunotherapy in acute leukemias extends beyond them offering hope for improved outcomes and expanded therapeutic options for patients.

## Figures and Tables

**Figure 1 cancers-15-04137-f001:**
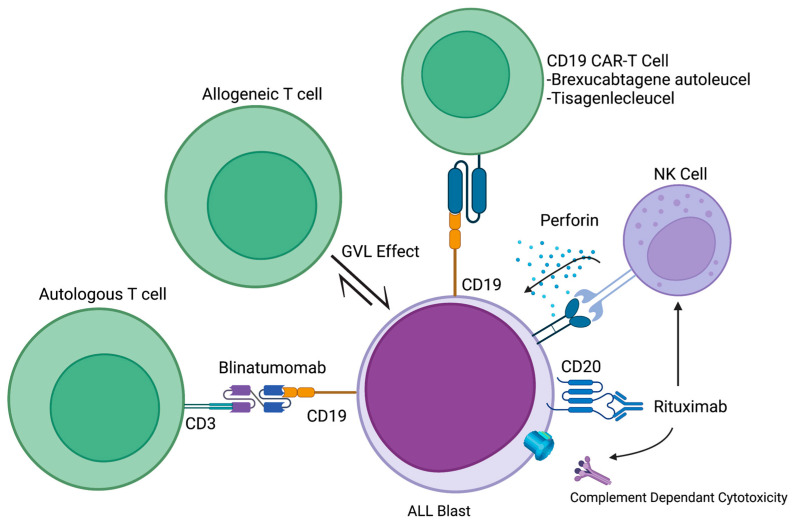
Select cells and signaling mechanisms involved in ALL immunotherapeutic targeting. Figure made with BioRender Scientific Image and Illustration Software.

**Figure 2 cancers-15-04137-f002:**
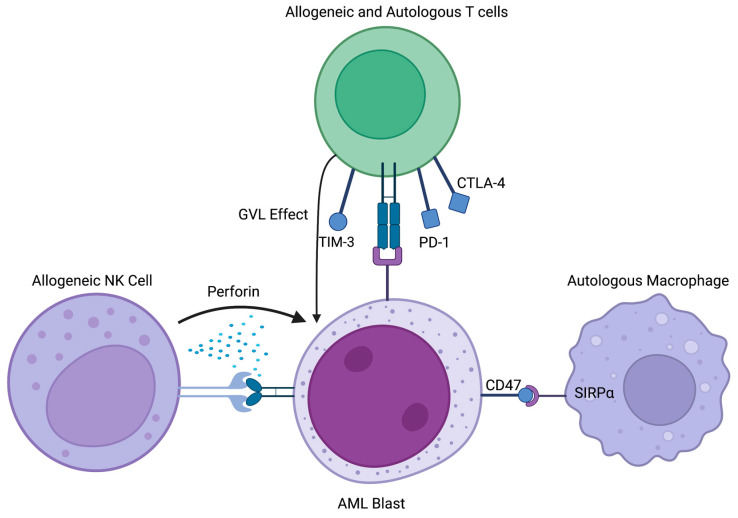
Select cells and signaling mechanisms involved in AML immunotherapeutic targeting. Figure made with BioRender Scientific Image and Illustration Software.

**Figure 3 cancers-15-04137-f003:**
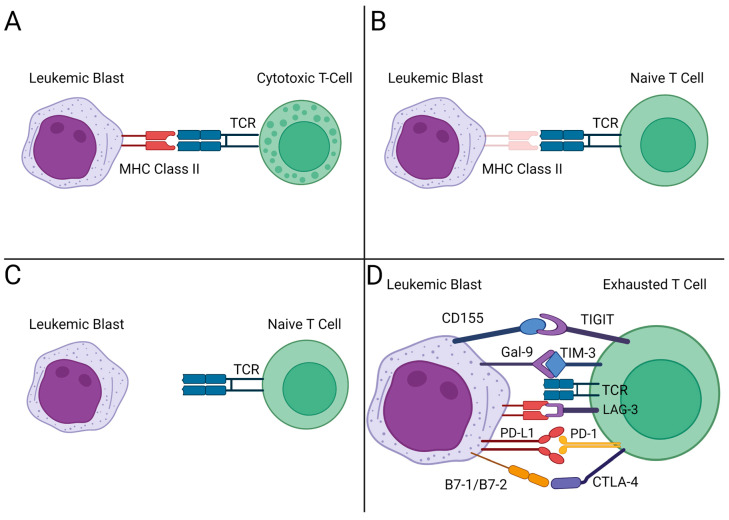
Immune escape in AML. (**A**) T-cells able to recognize leukemia blasts through MHC Class II-TCR signaling. (**B**) Downregulation of MHC Class II leads to decreased TCR activation. (**C**) Loss of MHC Class II leads to absent T-cell response. (**D**) Immune checkpoint upregulation by the leukemic blast causes immune escape. TCR: T-cell receptor. Figure made with BioRender Scientific Image and Illustration Software.

## Data Availability

There is no data to share.

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
