# Peer review of "Immunotherapy in Acute Leukemias: Past Success Paves the Way for Future Progress"

_cancers, 2023, doi:10.3390/cancers15164137_

Round 1

Reviewer 1 Report

Chergui and Reagen present a very succinct but very elegant review of the state of the art of immunotherapies in acute leukemias. The manuscript is well-written and illustrated. References are adequate.

Author Response

Reviewer 1 comments and response attached

Reviewer 2 Report

1) This manuscript reviews a difficult and complex topic but come up short on depth and overall the review is very superficial. In my opinion, adding in depth discussion of every section would greatly increase the quality of this review.

2) The review by Chergui and Reagan starts with an interesting historical description of the development of immunotherapy. However, the contributions of William B. Coley to the concept of immunotherapy have not been commented. I think that a brief reference to Coley's theory should be included.

3)This sentence in Page 3 is difficult to understand: "Previously the beneficial effect of allo-HSCT was thought to be primarily through the effects high doses of chemotherapy and/or radiation on residual host leukemia cells somewhat akin to the mechanisms at play in autologous hematopoietic stem cell transplant."

4) "Dependant" in Figure 1 should be "dependent".

5) The role of T cells in GVL effect in ALL compared to AML is not clearly explained.

6) PD-1 and CTLA-4 are mainly considered as inhibitory receptors (Page 3, section 3).

7) Page 4: "checkpoint" instead of "check point".

8) Table I must be completed with information about present clinical trials, including that of combination therapies.

9) Further uses of anti-CD20 must be discussed.

10) A section about the potential of NK  and CAR-NK cell therapies should be added.

The manuscript is, in general, well written.

Author Response

Reviewer 2 comments and response attached. 

Reviewer 3 Report

This is a well written review of the literature on immunotherapy in acute leukemias. Please see my comments/suggestions below:

1.     In the abstract the authors use the term ‘solid tumour patients’. Perhaps patients with solid tumours would be a more appropriate phrase to use.

2.     The figures appear to be generated using online tools for scientific illustration (such as Biorender). If this is the case, please acknowledge this as required by licensing requirements.

3.     It may be useful to have multiple figures describing each class of immunotherapy separately and in more detail. The current figures provide a good overview but could be more informative if expanded.

4.     Please provide a reference for the statement describing T-cell depletion, donor lymphocyte infusions, and post-transplant immunosuppression optimisation (Section2 paragraph 2 last line).

5.     Please cite references uniformly, sometimes a space is present between the word and the reference, sometimes it is before or after the period. There are several inconsistencies including reference 10, 12, 13, 14 ,1.

6.     After the description of TIM-3 (section3 paragraph 4) it would be useful to mention other checkpoint receptors such as TIGIT and LAG-3 which are potential candidates for blockade.

7.     On page 5 there is a mention of work done at Stanford University. It is not appropriate to name one institution (while not naming any others), this line should be removed.

8.     Phase II rather than phase 2 is the general convention while referring to clinical trials.

9.     The general convention is to use a superscript positive or negative sign while referring to expression of a marker on cells. Such as CD19+, CD8+, CD20+.

10.  It would be very beneficial to add in more commentary about relapse after immunotherapy, for example, CD19- clones have been detected in relapse after CD19-CAR therapy. Each section should include a section on relapse to provide a more balanced commentary highlighting the real-world challenges of immunotherapy, not just the benefits.

11.  The commentary on CRS with immunotherapy needs to be put into context. This would have been a major concern during early trials. However, it is currently well managed, generally without needing to stop therapy (as examples in the CAR-T and other settings). It is important to state that although CRS is a concern, it does not necessarily lead to treatment cessation.

Author Response

Reviewer 3 comments and response attached. 

Round 2

Reviewer 2 Report

The authors have satisfactorily answered all my comments.

The manuscript is well-written.